# Paths to adaptation under fluctuating nitrogen starvation: The spectrum of adaptive mutations in *Saccharomyces cerevisiae* is shaped by retrotransposons and microhomology-mediated recombination

**Michelle Hays**[1☯], **Katja Schwartz**[1☯], **Danica T. Schmidtke**[2], **Dimitra Aggeli**[1¤], **Gavin Sherlock**[1]*

**1** Department of Genetics, Stanford University School of Medicine, Stanford, California, United States of America, **2** Department of Microbiology and Immunology, Stanford University School of Medicine, Stanford, California, United States of America

☯ These authors contributed equally to this work.
¤ Current address: Department of Biological Sciences, Lehigh University, Bethlehem, Pennsylvania, United States of America
* gsherloc@stanford.edu

**Data Availability Statement:** Whole genome sequencing data and barcode sequencing data for

## Abstract

There are many mechanisms that give rise to genomic change: while point mutations are often emphasized in genomic analyses, evolution acts upon many other types of genetic changes that can result in less subtle perturbations. Changes in chromosome structure, DNA copy number, and novel transposon insertions all create large genomic changes, which can have correspondingly large impacts on phenotypes and fitness. In this study we investigate the spectrum of adaptive mutations that arise in a population under consistently fluctuating nitrogen conditions. We specifically contrast these adaptive alleles and the mutational mechanisms that create them, with mechanisms of adaptation under batch glucose limitation and constant selection in low, non-fluctuating nitrogen conditions to address if and how selection dynamics influence the molecular mechanisms of evolutionary adaptation. We observe that retrotransposon activity accounts for a substantial number of adaptive events, along with microhomology-mediated mechanisms of insertion, deletion, and gene conversion. In addition to loss of function alleles, which are often exploited in genetic screens, we identify putative gain of function alleles and alleles acting through as-of-yet unclear mechanisms. Taken together, our findings emphasize that *how* selection (fluctuating vs. non-fluctuating) is applied also shapes adaptation, just as the selective pressure (nitrogen vs. glucose) does itself. Fluctuating environments can activate different mutational mechanisms, shaping adaptive events accordingly. Experimental evolution, which allows a wider array of adaptive events to be assessed, is thus a complementary approach to both classical genetic screens and natural variation studies to characterize the genotype-to-phenotype-to-fitness map.

fitness remeasurement are available at the SRA under BioProject ID PRJNA882934.

**Funding:** Research reported in this publication was supported by the National Institute of Allergy and Infectious Diseases of the National Institutes of Health under Award Number F3 2AI160906 to MH. The content is solely the responsibility of the authors and does not necessarily represent the official views of the National Institutes of Health. MH was also supported by a Stanford Center for Computational Evolutionary and Human Genomics Postdoctoral fellowship, NIH NHGRI 5 T32 HG000044-24, and HHMI Hanna Gray (GT15996). DTS was supported by T32 GM00727646 and NSF GRFP DGE-1656518. The study was also funded by NIH grant R35 GM131824 to GS. The funders had no role in study design, data collection and analysis, decision to publish, or preparation of the manuscript.

**Competing interests:** The authors have declared that no competing interests exist.

## Author summary

A major driver of evolution is the occurrence and subsequent selection of adaptive mutations, which make an organism more fit in a given environment. While adaptive mutations are often point mutations, such as missense or nonsense mutations, there are many other types of mutation. Here we find that retrotransposition events, whereby a retrotransposon makes a new copy of itself elsewhere in the genome, are frequent when yeast are propagated by serial batch transfer in media with limiting nitrogen. Furthermore, we find many such events are adaptive, and show that the rate of retrotransposition is increased in nitrogen limited media compared to glucose limited media specifically when cells are propagated by serial batch transfer. However, we see no evidence of beneficial retrotransposition events when cells are propagated in nitrogen limited media in continuous culture. Our findings emphasize that *how* selection (fluctuating vs. non-fluctuating) is applied also shapes adaptation, just as the selective pressure (nitrogen vs. glucose) does itself.

## Introduction

Genomic changes often underlie phenotypic variation and population evolution. To understand phenotypic variation, genetic screens can identify relevant loci, but are often limited in the types of genomic changes they explore. For example, screens often use mutagens with specific mutational mechanisms, or collections of deletion mutants. Alternatively, comparative genomics can establish which changes have previously arisen and survived, but linking such changes to fitness and adaptation can be challenging. By contrast, experimental evolution can reveal how organisms adapt under specific selective pressures, revealing the most fit and/or frequent beneficial mutations (e.g. [1–6]). The mutational types that arise during experimental evolution are not limited to those typically isolated from genetic screens, but rather reflect the mutational mechanisms active in a population experiencing a selective pressure of interest. The relative rates and effects of different mutagenic mechanisms differ greatly in yeast, with single nucleotide polymorphisms (SNPs) arising at $\sim 10^{-10}$ per nucleotide per generation [7], while aneuploidy, for example, has been observed at $\sim 10^{-4}$ per generation [8] (though it can vary greatly between chromosomes and genetic backgrounds [9]) and affects hundreds of genes at a time. Other mutagenic mechanisms, such as retrotransposition, recombination-driven gene conversion, loss of heterozygosity, and structural rearrangements may also give rise to adaptive events unique to those mechanisms and can also occur during experimental evolution. Therefore, experimental evolution is an especially useful tool to understand how genome changes facilitate adaptation. By profiling the target loci at which adaptation occurs, the mutagenic mechanisms that create those changes, and the specific adaptive alleles that arise, agnostic to gain or loss of function, a more complete view of the genotype-to-phenotype map can emerge. Thus, experimental evolution complements, rather than simply reproduces, findings obtained through other types of genetic studies.

Some stress conditions are linked to increases in specific mutagenic mechanisms and stress-induced genomic instability may even improve evolvability of organisms [10]. Populations under stress may therefore sample more genotypic and phenotypic states, aiding population survival in environments for which ancestors were ill-adapted. Transposons and retrotransposons are mobilized in many species in response to stress (e.g. [11–20]), including in *Saccharomyces cerevisiae* [21–28], leading to an increase in mobile element-driven mutagenesis. *S. cerevisiae* has five families of Long Terminal Repeat (LTR) retrotransposons: Ty1—Ty5

[29,30]. Ty1 and Ty2 are closely related and are the two most abundant families in the reference genome. Ty1 transcription increases under DNA damaging conditions [27], extreme adenine starvation [23,24], filamentous growth (as a result of carbon or nitrogen starvation; [31]), and telomere shortening [26]. Newly integrated Ty copies can interrupt host gene coding sequences or alter host gene regulation, causing increased, decreased or even constitutive activation of neighboring ORFs. Indeed, others have shown that Ty1 includes a transcriptional enhancer that can increase expression of neighboring genes in the same orientation as the Ty1 element [32–35]. This regulatory capacity arguably makes retrotransposon mutagenesis more powerful for driving host genome evolution, and possibly even determining evolvability itself, than, for example, point mutations. Genomic variation also arises comparatively frequently through several microhomology mediated mechanisms [36]. These events, such as microhomology-mediated translocations, duplications, deletions, and gene conversion events, cause variants to arise more frequently specifically at sites of short sequences found tandemly repeated within a genome [36]. This increased frequency of site-specific mutation could influence population evolvability and adaptation, disproportionately impacting the evolution of genomic regions containing microhomology.

Laboratory-directed microbial evolution experiments allow investigation of how selective pressures shape genome evolution and most often make use of either serial transfer or continuous culture [37–39]. The selective pressure can come from nutrient limitation, the presence of an inhibitor (such as an antibiotic) or a restrictive environmental condition (e.g., temperature). In serial batch transfer evolutions, populations are iteratively passaged into fresh media, leading to regular, periodic environmental fluctuations. By contrast, in chemostat or turbidostat growth, continuous flow into the culture chamber of fresh medium and outflow of spent medium/cells produces a constant environment [40]. This fundamental difference between fluctuating vs. constant selective pressure can drive different means of genome evolution: the rates that some classes of mutations arise may differ between these approaches, and the relative fitness of evolved alleles may differ as well.

*S. cerevisiae's* adaptation to glucose limitation has been well-studied in both chemostat and serial transfer conditions and each results in different adaptive mutational spectra [3,6, 41–43]. Continuous culture in limiting glucose predominantly selects for mutants with more rapid glucose uptake, such as amplifications of the *HXT* glucose transporters [44], as well as lineages that fail to mount starvation stress responses [41]. By contrast, in fluctuating glucose conditions, self-diploidization and point mutations in the Ras and TOR pathways predominate [3], the latter often resulting in a shorter lag phase [45].

Like carbon, nitrogen is an essential nutrient for yeast survival. Nitrogen sensing and uptake also uses a complex signaling network with several feedback loops and points of self-regulation. Nitrogen-sensing impacts many other cell states as well, such as initiation of catabolic programs to scavenge nutrients from within and changes in growth programs such as filamentous growth and biofilm formation. Just as yeast favors glucose as a carbon source, it also preferentially favors different nitrogen sources (see [46]). Depending on environmental nitrogen sources available, yeast may import nitrogen through substrate-specific transporters (such as Mep1/3 and Mep2 for ammonia or Gnp1 for glutamine) or through more general transporters (such as the general amino acid permease—Gap1), or a combination thereof [47–50]. The nitrogen catabolite repression (NCR) and retrograde response pathways facilitate regulation and usage of different nitrogen sources (see [51,52]); both of these pathways involve TOR signaling, and NCR exhibits self-regulation. For example, Gat1 is a central regulator of NCR and participates in several feedback loops, including self-activation [53]. Gat1 is also required as a coactivator along with Gln3 at many NCR promoters. Additionally, NCR repressors Gzf3 and

Dal80 function by interfering with Gat1 binding (including competition at the *GAT1* promoter), *GAT1* expression and Gat1 cellular localization [54].

Previously, experimental evolution of *S. cerevisiae* under constant nitrogen limitation demonstrated adaptive amplification of transporter genes associated with the specific nitrogen sources available. For example, the *GAP1* transporter gene has been observed to amplify in glutamine-limited chemostats via several mechanisms, including extrachromosomal circle formation facilitated by recombination between flanking LTRs [55] and ODIRA [56,57]. *DUR3* is amplified under urea limitation, *PUT4* under proline limitation, *DAL4* under allantoin and *MEP2* under ammonium limitation [4,58]. Although the identities of the transporters amplified were dependent upon the nitrogen source available, permease amplification was frequently observed as a general class of adaptation. Conversely, loss of unutilized transporters can also be beneficial under continuous selection conditions [4,57].

We set out to determine if, as was observed in adaption to glucose limitation, different classes of mutational mechanisms give rise to different modes of adaptation under fluctuating nitrogen limitation as compared to constant limitation. The relative complexity of regulation, pathway feedback and pleiotropy of nitrogen signaling may give rise to a distinct adaptive spectrum under fluctuating conditions that vary between low and zero nitrogen during the growth cycle. This fluctuating nitrogen condition provides an interesting case for comparison of adaptive mutational spectra, contrasting adaptation to limiting nutrients (glucose vs. nitrogen), and also constant vs. fluctuating conditions. Under fluctuating nutrient availability, the limiting nutrient begins at a higher concentration than under constant selection conditions (e.g., ~6mM Nitrogen (3mM ammonium sulfate) in our study), but also drops to zero prior to cells being moved into fresh medium. By contrast, during continuous growth in a chemostat, the limited nutrient availability is held constant, but typically at a lower concentration (e.g., 0.8mM Nitrogen, regardless of the chemical form, in Hong *et al*). These comparisons provide insight into how the application of selection (constant vs. fluctuating nutrient availability) shapes available adaptive routes and their relative fitness, as well as how the genetic architecture of a trait (nitrogen vs. carbon sensing) influences the spectrum of beneficial mutations.

Previously, we analyzed the population dynamics of prototrophic *S. cerevisiae* evolved under nitrogen-limited conditions by serial transfer, with ammonium as the limiting source of nitrogen [59]. We noted an initial increase in adaptive single-mutants, followed by a crash in population diversity due to clonal interference as a result of rapid expansion of highly beneficial double-mutants. Here, we more deeply explore the specific mechanisms underlying adaptation to nitrogen limitation, focusing on recurrently mutated loci and the different mutational classes that give rise to these adaptive alleles. We identify six loci that are often targets of selection specifically under fluctuating nitrogen limitation where cells experience transient total nitrogen starvation. This suggests that mutation of these loci leads to the greatest fitness increases and/or that these loci are most easily and frequently mutated. In addition to SNPs and small indels, we observe several adaptive alleles that result from microhomology-mediated insertion, deletion or gene conversion, and multiple alleles that result from novel Ty1 or Ty2 insertions. This is in contrast to others' results under continuous growth conditions, where these classes of mutations are rarely observed and different loci are found most frequently mutated. While many of our identified adaptive alleles are likely loss of function mutants, we also identify putative gain of function mutations, again emphasizing the utility of experimental evolution for exploring the breadth and depth of adaptive alleles and the mechanisms that birth them.

## Results

### Summary of nutrient-limited evolution experiments and lineage tracking

We previously evolved yeast by serial transfer under both glucose and nitrogen limitation [59,60], using lineage tracking by barcode sequencing to identify adaptive lineages and to estimate fitness. Evolved clones were previously isolated for fitness validation and whole genome sequencing; briefly, ~5,000 clones were isolated from generations 88 and 192 (from glucose and nitrogen limitation respectively), their barcodes identified, and their ploidy was determined using a benomyl based assay as described ([3]; see Methods).

### Sequencing of evolved clones from nitrogen limitation

To identify the underlying adaptive mutations, we whole-genome sequenced 345 clones with unique barcodes isolated from generation 192 of the nitrogen-limited evolution. These clones were selected based on the previously characterized barcode frequencies and their estimated fitness during the evolution itself and were expected to include both neutral and adaptive clones. It is known that autodiploidization can confer ~0.03 fitness increase [3,59]. For diploids in this study, we only whole genome sequenced clones with estimated fitness greater than autodiploidy alone would provide (>0.03), assuming these diploids would harbor additional beneficial mutations that account for the additional fitness increase. We analyzed the data for SNPs, small indels, and *de novo* retrotransposition events; sequenced clones contained from zero to five SNPs/indels and zero to nine Ty1/2 insertions (see Methods). To determine which of the observed mutations are likely adaptive, we looked for genomic loci that were recurrently mutated and whether these mutations correlated with reproducible fitness increases by fitness remeasurement (see below; Table 1). Notably, we identified four loci that were recurrently mutated several times each (*GAT1*, *PAR32*, *MEP1* and *FCY2* with 73, 45, 44 and 27 unique adaptive mutations respectively). In addition, we also recovered mutations with increased fitness in nine other loci, but with four or fewer unique adaptive mutations at each.

### Fitness of evolved clones in both limiting nitrogen and glucose

To validate the increased fitness of unique isolated clones that were whole genome sequenced, fitness remeasurements were performed via pooled competitive fitness assay (Methods) in both limiting glucose and limiting nitrogen conditions. This allowed us to distinguish between lineages specifically adaptive in limiting nitrogen, and lineages adaptive in both conditions. The barcoded clones were pooled with 48 confirmed neutral strains from previous experiments [3] and were competed 1:10 against an unbarcoded ancestor in triplicate. Lineage trajectories were highly reproducible (S1 Fig) as were fitnesses calculated from those trajectories (S2 Fig).

Many of the clones isolated from the nitrogen-limited evolution were beneficial only under nitrogen-limited conditions (332/345). The remaining clones were neutral or even maladaptive in glucose limitation (Fig 1, upper left region), or found to be roughly equally beneficial under both glucose limitation and nitrogen limitation (Fig 1, upper right). For the remainder of this manuscript, we focus on genes whose mutation provides substantially greater fitness under nitrogen limitation than under glucose limitation.

### Identification of presumptive beneficial mutations

To classify mutations as specifically adaptive in nitrogen limitation, we identified loci that met three criteria: 1) were recurrently mutated in independent lineages, 2) were validated to a fitness effect of >0.01 in nitrogen-limited media (see Methods), and 3) have substantially greater

**Table 1. Genes with recurrent mutations identified in both haploids and diploids, or whose mutation has been previously shown to be adaptive.** Mutations in bolded genes are specifically "nitrogen adaptive" (see below). Ty1 and Ty2 were the only retrotransposons responsible for novel adaptive insertions, and nearly all new insertions are Ty1. For additional details on specific Ty1/2 insertions such as location, family and orientation, see S2 and S4 Files.

| | Number of alleles | | | | | | | | | | | |
| | SNPs | | | | Microhomology mediated mutations | | | Ty1/2 insertions | | | | |
| Gene | Missense | Nonsense | Indels | Other SNPs | MHMI | MHMD | MHMGC | Ty in ORF | 5' Ty | 3' Ty | Other Ty related rearrangement | Whole gene deletion |
|---|---|---|---|---|---|---|---|---|---|---|---|---|
| *GAT1* | 12 | 2 | 2 | 1 | 1 | 4 | 0 | 47 | 3 | 0 | 0 | 1 |
| *PAR32* | 1 | 6 | 3 | 0 | 3 | 2 | 0 | 23 | 5 | 0 | 2 | 0 |
| *MEP1* | 8 | 0 | 0 | 0 | 0 | 0 | 0 | 8 | 0 | 28 | 0 | 0 |
| *MEP2* | 2 | 0 | 0 | 0 | 0 | 0 | 0 | 0 | 0 | 0 | 0 | 0 |
| *MEP3* | 2 | 0 | 0 | 0 | 0 | 0 | 0 | 0 | 0 | 0 | 0 | 0 |
| *FCY2* | 4 | 4 | 2 | 0 | 0 | 1 | 10 | 5 | 1 | 0 | 0 | 0 |
| *ARO80* | 3 | 0 | 0 | 0 | 0 | 0 | 0 | 1 | 0 | 0 | 0 | 0 |
| *TOR1* | 1 | 0 | 0 | 0 | 0 | 0 | 0 | 0 | 0 | 0 | 0 | 0 |
| *TOR2* | 0 | 1 | 0 | 0 | 0 | 0 | 0 | 0 | 0 | 0 | 0 | 0 |
| *GPB1* | 0 | 0 | 0 | 0 | 0 | 0 | 0 | 1 | 0 | 0 | 0 | 0 |
| *GPB2* | 2 | 0 | 0 | 0 | 0 | 0 | 0 | 0 | 0 | 0 | 0 | 0 |
| *SSK2* | 1 | 0 | 0 | 0 | 0 | 0 | 0 | 0 | 0 | 0 | 0 | 0 |
| *PDE1* | 1 | 0 | 0 | 0 | 0 | 0 | 0 | 0 | 0 | 0 | 0 | 0 |

increased fitness in nitrogen limitation than in glucose limitation. Independent lineages were designated by the presence of unique barcodes. We identified six loci that satisfied these criteria, which we refer to as "nitrogen adaptive" from here forward: *GAT1*, *FCY2*, *PAR32*, *MEP1*, *MEP2*, and *MEP3* (Table 1). Together, these loci were independently mutated a total of 193 times across multiple lineages that exhibited reproducible fitness increases. *MEP2* and *MEP3* were only observed as mutated twice each, with one of the *MEP2* mutations occurring heterozygously in a diploid, while we identified 44 adaptive *MEP1* mutations. Notably, and in contrast to prior results from chemostat evolutions, we did not observe recurrent amplifications at the *MEP* loci, the permeases associated with ammonium import, the limited nitrogen source in this experiment. We identified an additional six loci that had also acquired multiple independent mutations during our evolution; however, they did not show reproducible fitness gains, making the status of these mutations (adaptive or neutral) inconclusive: *MIT1*, *DAL81*, *VID28*, *PFA4*, *CRP1*, and *FAS1*. While *FAS1* was found mutated three times independently in confirmed adaptive lineages, these mutations were only identified in the context of *PAR32* or *GAT1* mutant backgrounds and seemingly did not increase fitness beyond that of the confirmed adaptive alleles. Finally, some loci were mutated and adaptive in both glucose- and nitrogen-limited conditions, including *ARO80* and members of the TOR and RAS pathways: *TOR1*, *TOR2*, *LST8*, *SCH9*, and *GPB1* and *GPB2* (Table 1 and S1 File).

## Fitness gains and validation of nitrogen adaptive loci

The pooled fitness assays showed that the six recurrently mutated loci consistently correlate with increased fitness in nitrogen limitation, and moreover that different alleles show consistent fitness increases (Fig 1). For example, most of *GAT1* mutations show a reproducible fitness increase of 0.04–0.06/generation, while *MEP1* mutations fall in two clusters with Ty1/2 insertions having an average fitness increase of 0.03–0.05 and missense mutations affecting residue Asp262 have a fitness increase of ~0.08. The *MEP2* and *MEP3* alleles showed fitness gains between 0.04 and 0.07. Together, these data show that the evolved alleles at these six loci

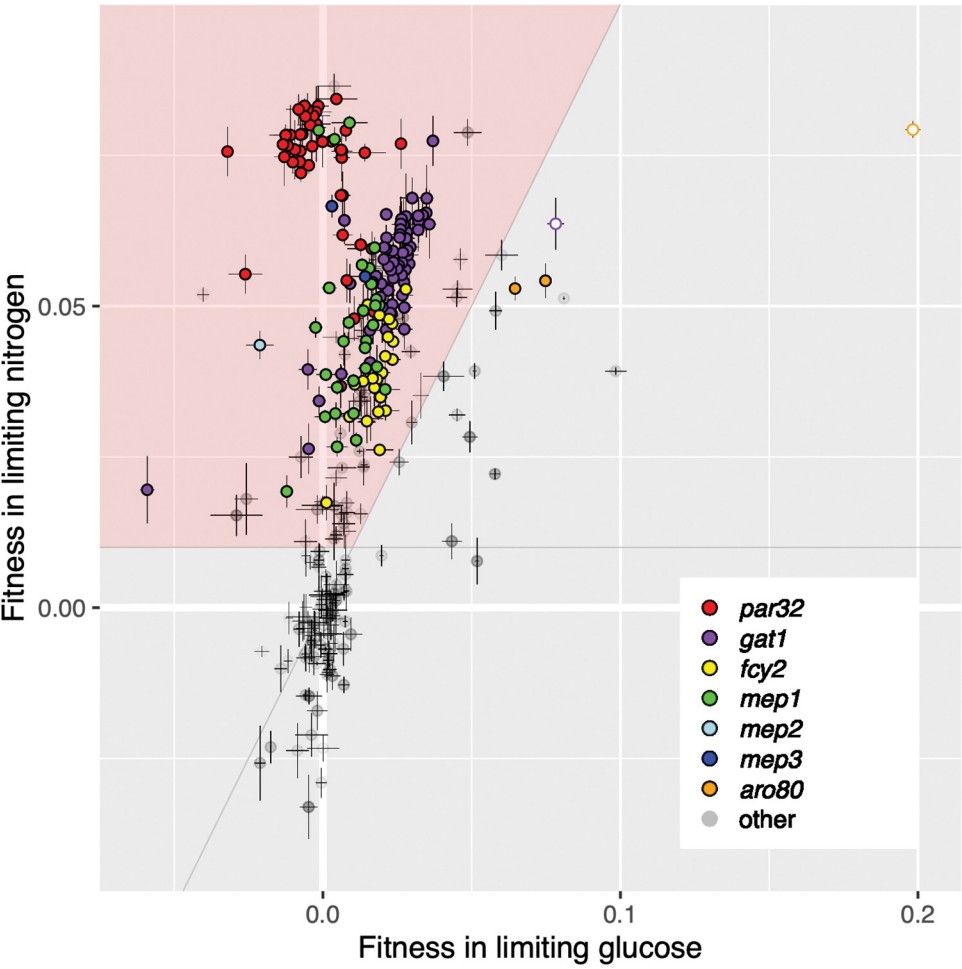

**Fig 1. Fitness remeasurements of haploid clones isolated from the nitrogen-limited evolution in nitrogen versus glucose limitation.** Clones in the upper left region (pink) are considered "nitrogen-adaptive" (see text). The two white-filled circles correspond to double mutants, a *gat1* mutant with a *ssk2* mutation, and an *aro80* mutant with a *gpb2* mutation. Error bars indicate standard deviation of the replicate fitness measurements.

are indeed adaptive mutations, reproducibly driving 0.03–0.08 increases in fitness in fluctuating nitrogen starvation conditions. To further confirm these mutations are indeed the drivers of fitness gains (rather than 'hitchhiking' neutral mutations), we generated progeny for one or more alleles of each gene for five of the nitrogen adaptive loci and performed a pooled competitive fitness assay as above. The resulting fitness estimates confirmed that the observed fitness gains are correlated with mutations in *GAT1*, *MEP1*, *MEP2*, *MEP3, and PAR32*. Progeny show equivalent fitness gains to the adaptive parent, demonstrating that mutations at these loci likely account for all, or nearly all, of the fitness gain in these lineages (Fig 2). Note, the magnitudes of the measured fitnesses are often somewhat smaller than in the fitness remeasurements in Fig 1; we hypothesize that this is a result of the competing pool composition being different (see Methods).

## Targets of adaptation

Lineages with mutations in the six nitrogen-adaptive loci account for 76% of adaptive lineages sequenced in our evolution (178/224 adaptive haploid lineages and 15/29 adaptive diploid

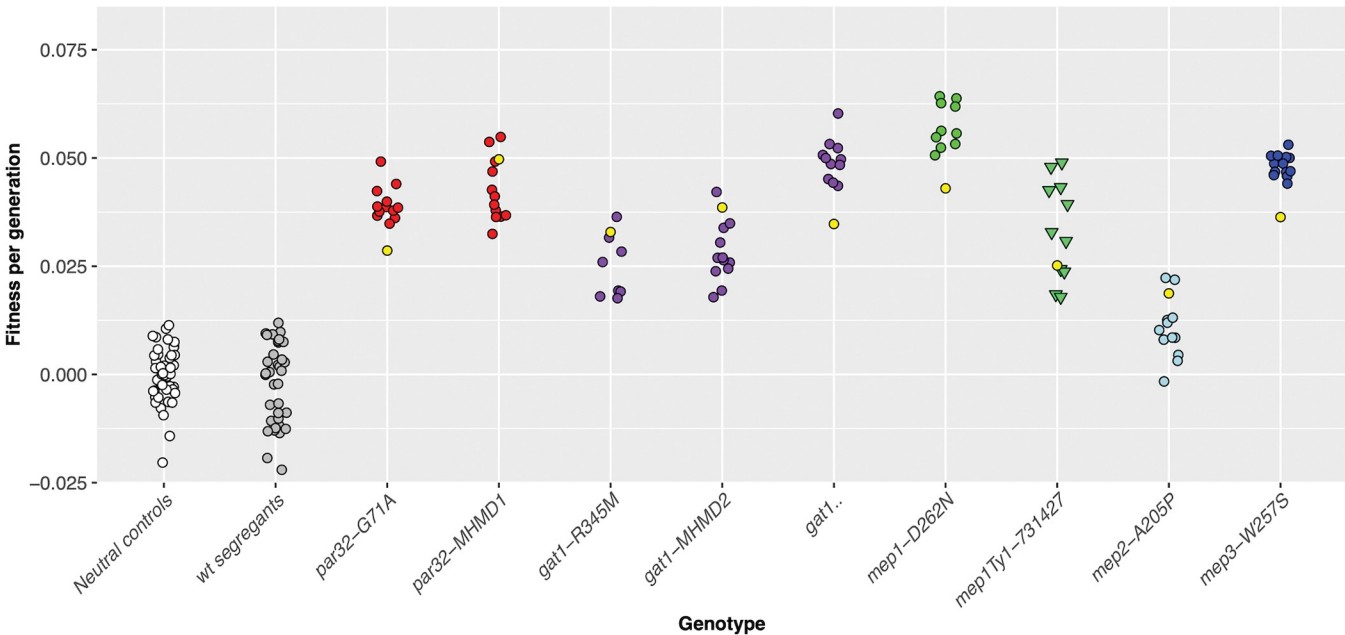

**Fig 2. Fitness remeasurements of progeny containing putatively adaptive alleles confirm adaptive mutations.** Neutral controls are the neutral lineages previously isolated from glucose-limited evolutions, while wild-type segregants are segregants from the crosses that lack the adaptive alleles of interest, but include passenger mutations. Yellow dots correspond to the parental lineage fitness remeasurements, while white dots outlined in red indicate the median. Wilcoxon rank-sum tests indicated all sample sets above are significantly different from neutral controls at p<0.0001 with the exception of the WT segregants which are not significant; *gat1*-MHMD2 p = 1.40E-12, *gat1*-R345M p = 1.40E-09, *gat1*Δ p = 7.10E-12, *mep1*-D262N p = 3.80E-11, *mep1*Ty-731427 p = 7.10E-12, *mep2*-A205P p = 1.40E-05, *mep3*-W257S p = 1.60E-14, *par32*-G71A p = 1.40E-12, *par32*-MHMD1 p = 1.40E-12, WT segregants p = 0.47 (n.s.).

lineages). Here we further discuss the mutation types and allele locations observed, and their roles as adaptive targets (Fig 3). We note that in some cases, the type of mutation observed is less indicative of fitness impact than the locus at which the mutation occurs. For example, mutations affecting the *FCY2* locus generally result in more modest fitness increases than mutations in *PAR32*, irrespective of mutation type.

## *GAT1* locus

*GAT1* encodes a GATA-type DNA-binding zinc-finger transcription factor which, among other activities, activates nitrogen catabolite repression (NCR) genes such as *GAP1* and *GLN1*. Gat1 binds DNA directly, but also facilitates transcriptional regulation via protein-protein interactions with other transcription factors [54]. *GAT1* transcription itself is typically upregulated by Gln3 under nitrogen-limited conditions, reducing nitrogen utilization within the cell. When a preferred nitrogen source is available, Gat1 and Gln3 are phosphorylated by TOR kinases and are sequestered to the cytoplasm, while *GAT1* transcription is simultaneously repressed by Ure2 and Dal80.

At the *GAT1* locus, novel Ty1/2 insertions account for a large proportion (68%) of adaptive mutations (50/73) in this evolution, predominantly inserting in the 5' half of the CDS (Table 1 and Fig 3A). Only three of the novel insertions arose from Ty2, with the remainder coming from the Ty1 family (S2 File). However, other classes of mutations were also observed in *GAT1*, including two nonsense mutations, two frameshift mutations, a lost stop codon, several short, out-of-frame intragenic rearrangements that likely arose by microhomology mediated recombination (Table 1 and Figs 3A and 4B), and 12 missense mutations, as well as a whole-locus deletion (3.9 kb, from 722bp upstream to 1.6kb downstream).

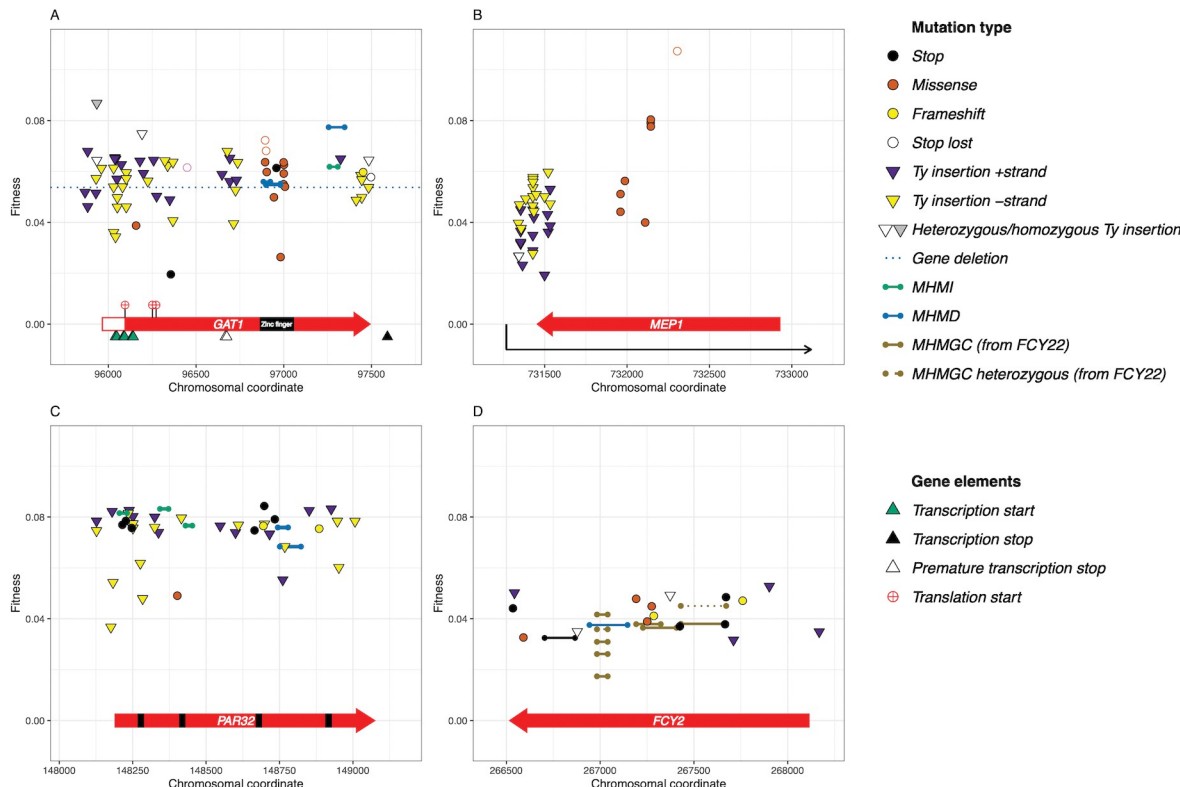

**Fig 3. Targets of adaptation and their fitness effects.** A) *GAT1*, B) *MEP1*, C) *PAR32*, D) *FCY2* loci and associated adaptive mutations. Glyphs indicate mutation type (see key); the dotted line for *GAT1* indicates the fitness of a complete deletion mutant; the black boxes in *PAR32* indicate repeat regions.

Several (8/29) adaptive diploid clones were isolated bearing *GAT1* mutations (open shapes, Fig 3A) in addition to the 65 haploids. Diploidization alone gives rise to increased fitness of ~0.03, while diploids with heterozygous missense mutations at *GAT1* experience fitness increases of 0.06–0.07, similar to *GAT1* mutant haploids. Homozygous *GAT1* mutants had even larger fitness gains. For example, a single diploid with a homozygous *GAT1* mutation (Ty insertion, putative null, also observed in a haploid background) was identified for which we measured a fitness increase of greater than 0.08. Additionally, several of these adaptive mutations putatively affect the *GAT1* RNA isoforms differentially (triangles below gene name), either through impacting alternate start (green triangle) or alternate stop sites (white and black triangles). Across both haploid and diploid clones, all but one (11/12) of the missense mutations affecting *GAT1* (Fig 3A, burgundy circles) fall within the zinc-finger domain and show fitness increases of ~0.06, similar to the putative null mutant alleles [61,62]. We also identified two alleles with small deletions in the zinc finger domain that each result in a frameshift that truncates Gat1.

In our study, we observe adaptive *GAT1* alleles with polymorphisms at the DNA-binding site, as well as putative null mutations. These adaptive alleles all show similar fitness increases (0.06±0.01) in haploid isolates. Given the many mechanisms by which Gat1 affect NCR regulation, these alleles may act on cell fitness through similar or differing mechanisms: Gat1 DNA binding mutants maybe functionally null, explaining the similarity in fitness increase, or perhaps DNA-binding mutants impact fitness through alternative means, such as modulating Gat1p protein-protein interactions with other transcription factors.

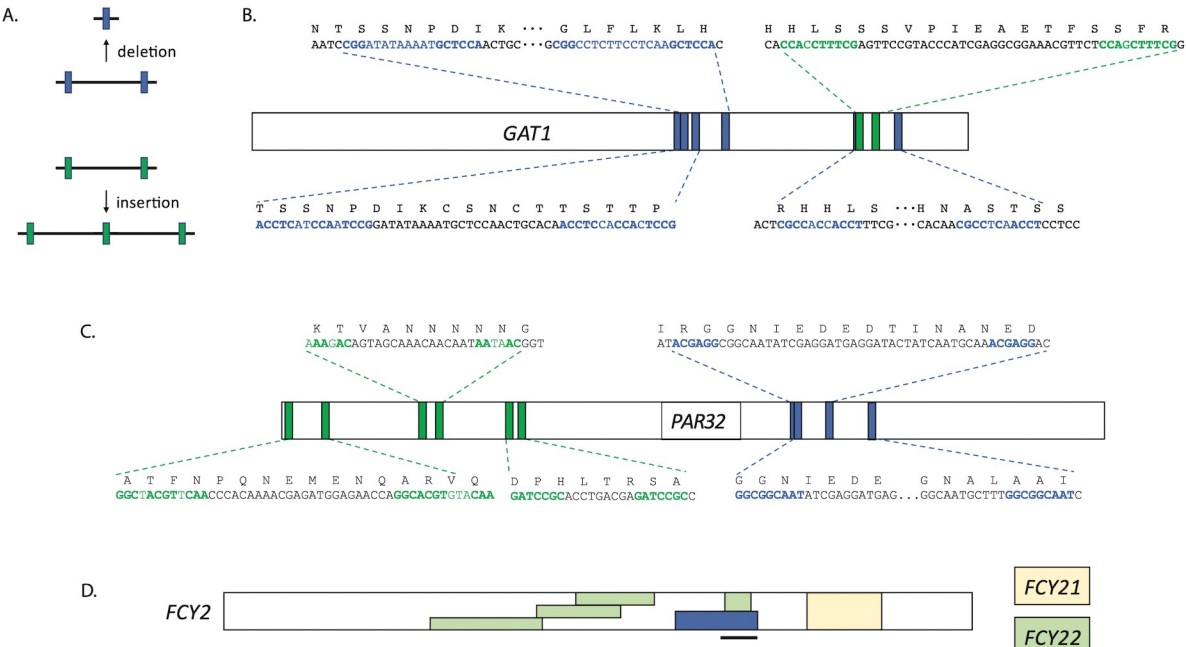

**Fig 4. Microhomology-mediated mutations.** A) diagram of how sequence flanked by regions of microhomology can either be deleted or duplicated; color is used to denote insertion vs. deletion. B) microhomology mediated mutations in *GAT1*, C) microhomology mediated mutations in *PAR32*, and D) regions of *FCY2* replaced by ectopic recombination from either *FCY21* or *FCY22*; one of the *FCY22* gene conversions was observed five independent times. The blue box in *FCY2* indicates the location of a microhomology mediated deletion, while the black bar indicates the location of previously observed gene conversion events in *FCY2* [74].

The NCR program allows cells to increase their uptake and usage of non-preferred nitrogen sources during low nitrogen abundance but is repressed in the presence of preferred nitrogen sources. *GAT1* null mutations may be beneficial under these evolution conditions (serial transfer in ammonium-limited media) as *gat1* cells may be better poised to immediately utilize the nitrogen available upon transfer to new media, relative to *GAT1* competitors that would require repression of NCR factors to utilize the freshly introduced nitrogen supply. For this reason, *gat1* null alleles may be most likely to be beneficial in cells experiencing fluctuating ammonium concentrations from moderate (3 mM) to total nitrogen starvation. While *GAT1* alleles have also been shown to be beneficial in constant low nitrogen chemostat environments [63,64], only adaptive DNA-binding SNP mutations were identified in that study. No null alleles were observed in adaptive clones or populations evolved in chemostat low ammonium conditions, even in ultradeep targeted sequencing experiments [4]. When *gat1* null allele fitness was assessed in those conditions they did not observe increased fitness [64].

## *MEP* loci

*S. cerevisiae* has three paralogs, *MEP1*, *MEP2*, *and MEP3*, that encode permeases for ammonium, the nitrogen source used in this study; these transporters have differing affinities for ammonium and are induced at different levels in response to the quality and quantity of the nitrogen source [65]. Mep1 forms a heterotrimeric complex with Mep3, while Mep2 forms a homotrimeric complex (see [66]). The Mep1/3 complex is inactivated by Par32 binding ([66]; see below), while Mep2 is regulated directly via Npr1 phosphorylation [67]. In our evolution conditions, we observe 44 independent mutations affecting the *MEP1* locus, and just two mutations each at *MEP2* and *MEP3* loci (Figs 3B and S3 and Table 1). Fitness effects of the

*MEP2* and *MEP3* mutations were comparable to some missense *MEP1* mutants (~0.04–0.06), while the fittest *MEP1* mutants conferred up to 0.08 advantage in haploids.

Notably, we found no evidence for amplification of the *MEP* transporters in adaptive clones, only the missense and Ty1/2 derived alleles described here. This is in contrast to the amplification of genes encoding relevant permeases observed under constant nitrogen limitation in chemostats [4,58]. This observation supports that how selection is applied (e.g., fluctuating vs. constant) under nitrogen limitation changes the spectrum of adaptive mutations recovered, as was observed under glucose limitation.

We observed four independent missense mutations that affected Asp261/262 in *MEP3* and *MEP1*, respectively. Strikingly, two of these resulted in an identical substitution in *MEP1* and *MEP3*, from Asp to Asn (S3A Fig). Others have observed *MEP* mutations under nitrogen-limited chemostat evolution conditions, interestingly only in *MEP2*, not in *MEP1/3* [4]. It is possible that the missense mutations at these sites give rise to a higher affinity ammonium transporter in all cases, but that differences in constant vs. fluctuating nitrogen-limited environments determine which transporter is most important or active.

In addition to missense mutations, we observed a striking pattern of Ty1/2 insertions at the *MEP1* locus (Fig 3B), with 36 novel insertions falling in the 3' UTR and the 3' most region of the gene. As with other loci, Ty1 activity accounts for the majority of new insertions, with just three Ty2 insertions at the *MEP1* locus (S2 File). These insertions fall close to the 5' end of an antisense transcript (SUT128) that has been observed in RNA-Seq experiments ([68]; S3B Fig). The group of Ty1/2 insertions in the 3' region unique to the *MEP1* locus may affect fitness in nitrogen limitation via a mechanism different than the putative gain of function missense mutations in the coding region itself. These alleles may be increasing Mep1p activity (via RNA regulation for example) or affecting NCR regulation independently of Mep1p. *MEP1* expression is normally controlled by nitrogen catabolite repression regulation via TORC1p and Par32p [66]. The lack of such insertions downstream of either *MEP2* or *MEP3*, and the fact that there is no annotated antisense stable unannotated transcript for either of them [68], suggests that this mode of regulation may be unique to *MEP1*.

Because serial transfer in ammonium-limited conditions gave rise to two main classes of adaptive mutation in *MEP1* (missense mutations in the transport domain, and 3' Ty1/2 insertions) and similar missense mutations in *MEP2* were observed in chemostat ammonium-limited conditions, we asked whether continuous culture conditions also give rise to Ty insertions near *MEP* genes. We reanalyzed the data from [4] (see Methods) to look for Ty insertions at *MEP1/2/3* and found no evidence for adaptive Ty insertions at these loci. This suggests that 1) Ty elements are less active under continuous culture conditions (even when the same nutrient is limiting), or 2) that the mechanism by which Ty insertions are beneficial under fluctuating nitrogen starvation conditions is not equally adaptive in constant low-nitrogen conditions, or 3) a combination of both. We speculate that the adaptive *MEP1* missense alleles observed in our study may in fact be gain of function mutants, capable of transporting ammonium more effectively, while the downstream Ty1/2 insertions may increase Mep1 activity through other means.

### *PAR32* locus

We observed 45 independent beneficial *PAR32* mutations, predominantly comprised of putative null alleles [44/45], including nonsense mutations, Ty1/2 insertions or frameshift mutations (Fig 3C), and just a single adaptive missense allele. Five frameshift alleles all likely arose by microhomology-mediated recombination (Fig 4C). The Ty-derived alleles were primarily due to Ty1 activity, with just 4/28 novel insertions arising from Ty2 activity (S2 File). Most of

the putative null alleles are associated with a ~0.08 fitness increase, suggesting that lack of Par32 gives rise to similar host fitness benefits regardless of mutation type that creates the null allele. Par32 is an unstructured protein known to inhibit Mep1/Mep3 transporters, so *par32* null alleles likely affect fitness through increasing Mep activity, just as *MEP* loci gain of function alleles would be beneficial in these same evolution conditions [66]. Consistent with this, the *par32* alleles fitness effect is similar to the effect of the fittest *mep1* alleles (~0.08). We observed only one *PAR32* missense mutation (Fig 3C, burgundy circle). This missense mutation lands in the second of four repeat motifs (Fig 3C, black bars) found in Par32; these repeat regions are essential for direct Par32 interaction with the Mep proteins [66].

Null *par32* alleles are likely to be adaptive by preventing the inactivation of Mep1/3 transporters specifically during fluctuating low-to-no nitrogen conditions. *par32* null alleles may facilitate more rapid uptake of nitrogen when moved to fresh medium containing ammonium relative to wild type cells that would need to re-activate Mep1/3 expression. Under continuous culture conditions where low ammonium concentrations remain steady and do not drop to zero, others have observed that *par32* null mutants are actually maladaptive [63].

### *FCY2* locus

Finally, we found the *FCY2* locus mutated in 27 lineages in our evolution (Fig 3D). *FCY2* is a purine-cytosine permease found at the plasma membrane, important for the import of adenine and other substrates into the cell [69–73]. Ten out of the 27 *FCY2* beneficial mutations are microhomology-mediated gene conversions with regions of other *FCY* genes (*FCY21* and *FYC22*; Fig 4D) resulting in frameshifts. Fcy21 and 22 are both nucleobase transmembrane transporters, but cannot functionally replace Fcy2 [69]. Such gene conversion events have been observed previously between *FCY2* and *FCY22*, but not involving *FCY21* [74]. These regions of gene conversion likely decrease or abrogate the transport function of Fcy2 as was seen previously [74]. The *FCY21/22* converted alleles show average fitness increases of ~0.04.

Additionally, we observed several Ty1 insertions in the 5' region of the CDS of *FCY2* (Fig 3D and Table 1) as well as both nonsense and frameshift mutations, suggesting that null alleles are also beneficial under nitrogen starvation. These mutations exhibit a similar range of fitness increases, again suggesting that a decrease in *FCY2* activity is likely adaptive in ammonium-limited serial transfer.

## Timing and establishment of adaptive mutations

Through barcode lineage tracking and comparison to neutral fitness lineages, we explored the relative dynamics of adaptive allele establishment in the population (S4 Fig). We note that the lower fitness, but still adaptive, alleles (such as mutations of *fcy2*, yellow S4E Fig) established early in the evolution, while single mutations corresponding to larger fitness increases (e.g., *PAR32* and *GAT1*, red (S4A Fig) and purple (S4B Fig) respectively) increased more rapidly but arose later in the evolution. These observations may reflect the 'ease' of acquiring these mutations: for example, *fcy2* appears to be the result of a gene conversion event, which may arise more often than inactivating point mutations in *par32 and gat1*, or the putative gain of function mutants in *MEP* genes. In addition to how frequently some mutation types occur, the relative fitness of a mutation shapes how quickly those adaptive lineages expand in the population. As we have observed previously [60], lower fitness adaptive mutations arise early during evolution, while higher fitness single mutations can occur later and still reach population frequencies high enough to be sampled in our clonal sequencing (S4E Fig).

## Mutational spectra in nitrogen- vs. glucose-limited populations: Increased Ty1/2 retrotransposition in nitrogen-limited conditions

To better understand the classes of adaptive mutations underlying genome evolution, we compared the mutational spectra observed in both neutral and adaptive populations under both nitrogen and glucose limitation. First, we note that autodiploidy itself confers a significant fitness benefit in both nitrogen limitation [59] and glucose limitation [3]. Beyond autodiploidy, we observed that in both neutral and adaptive haploids, SNPs comprise the majority of genome changes under glucose limitation (Fig 5A) and in both glucose and nitrogen-limited conditions SNPs were enriched in adaptive lineages relative to neutral lineages (black bars, Fig 5A). By contrast, in our haploid clones, Ty1/2 insertions are most prevalent under nitrogen limitation, with approximately ~2.6 novel insertions per clone compared to only ~0.47 novel Ty insertions in haploid clones from glucose-limited evolutions (Fig 5A). Novel Ty insertions were more prevalent in neutral than adaptive lineages (blue bars), however in nitrogen limitation in particular, Ty mutagenesis accounts for a substantial number of adaptive mutations (gold bar). In 345 nitrogen-evolved clones (which includes both haploid and diploid isolates) we observed 898 novel Ty insertions, while we found only 212 novel Ty insertions in 422 glucose-evolved clones (Fig 5B).

## Not all Ty families are equally activated under nitrogen limitation

To better understand how nitrogen limitation increases retrotransposition, we examined which Ty elements in the ancestral strain were most active under these conditions as compared to under glucose limitation (Fig 5B). Ty1 and Ty2 are the most abundant classes of Ty element

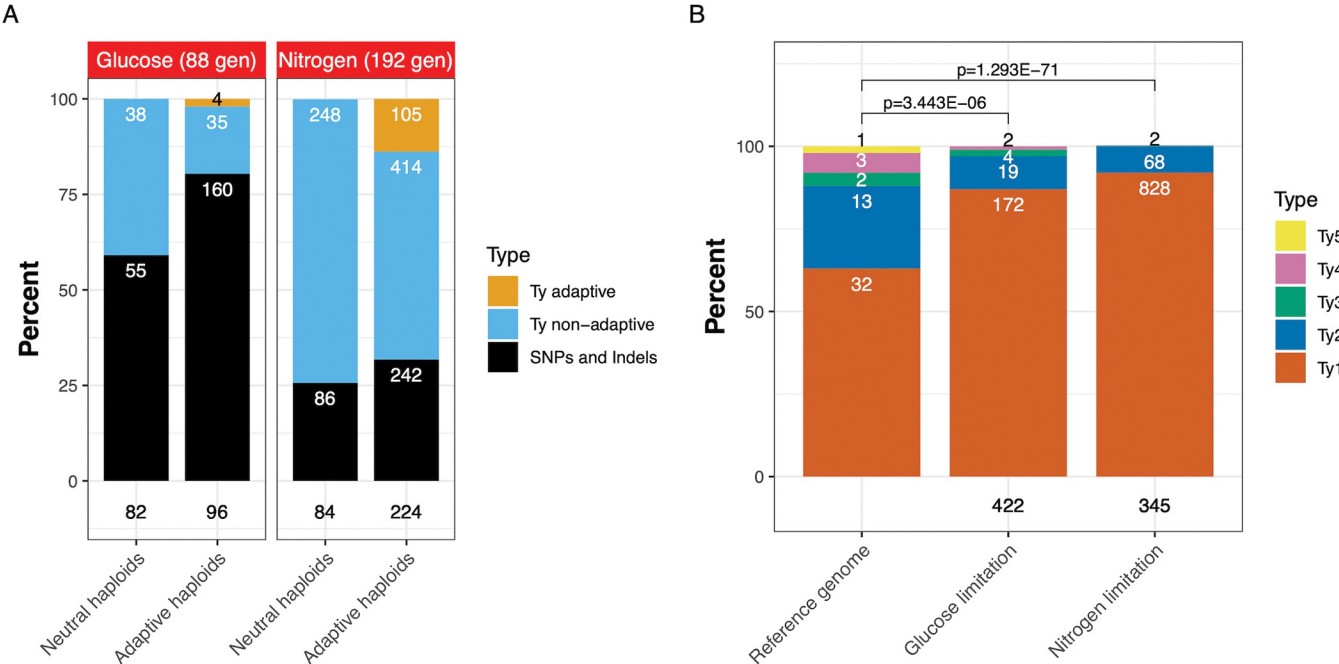

**Fig 5. Ty1/2 activity is increased in fluctuating nitrogen-limited conditions and gives rise to a significant number of adaptive alleles.** (A) Observed mutational types in haploid clones contrasting neutral vs. adaptive mutations isolated under glucose and nitrogen limitation. (B) Ty activity in both haploids and diploids evolved under nitrogen and glucose-limited conditions relative to the ancestral strain (reference genome). Stacked bar charts represent the fraction of each Ty type (Ty1-Ty5) present in the reference genome, or of novel insertions identified in clones evolved in the nitrogen- and glucose-limited conditions. Numbers on bars represent raw counts of the given Ty type. Numbers below the stacked bars indicate the number of strains from which the novel insertions were observed. P-values calculated by chi-square test.

[75] and are highly active in yeast, and the novel insertions observed in both our glucose- and nitrogen-limited evolved strains reflect that (Fig 5B). Ty1 is the most prevalent class in the ancestral strain (accounting for 32/50 intact Ty insertions). In nitrogen limitation, Ty1 accounts for 92% (828/898) of novel insertions with Ty2 accounting for just the remaining 8% (68/898), as opposed to the 26% of insertions in the ancestor. Ty1 is also frequently the most activated Ty element in *S. cerevisiae* under other stress conditions [31,76]. In glucose-limited clones, Ty1 accounts for 87% of novel insertions (172/197), while Ty2 accounts for ~10% (19/197) of novel insertions. In the ancestor, Ty3 and Ty4 account for 4% and 6% of insertions respectively, but just 0.2% (2/898) of Ty3 in nitrogen-limited clones. The single copy of Ty5 in the ancestor is known to be inactive and gives rise to no new insertions under either glucose or nitrogen limitation [77]. The distributions of novel Ty insertions by Ty types are significantly different in nitrogen- and glucose-limited evolved clones compared to the distribution of Ty insertions in the reference genome (chi-squared p-value = 1.3e-71 and 1.4e-10, respectively), suggesting, as has been seen previously, that not all Tys or Ty classes are equally active. The distributions of Ty classes in the set of novel insertions those clones evolved in nitrogen and glucose also differ significantly (p = 3.4e6).

Taken together these data suggest that while Ty activity is substantially increased under fluctuating nitrogen conditions (as compared to both glucose limitation and constant nitrogen limitation), that not all Tys are equally activated, with Ty1 and Ty2 being the predominant *de novo* insertions. Because the number of novel Ty1/2 insertions is substantially greater in nitrogen-limited clones as compared to glucose-limited clones, but this increase is seen in both neutral as well as adaptive clones (Fig 5A), we infer that this condition-specific difference is at the level of Ty1/2 activity rather than at the level of selection (blue bars). Indeed, even within adaptive clones, only a fraction of the novel Ty1/2 insertions are driving the fitness increase (gold bars) while other Ty1/2 insertions are likely passenger mutations (blue bars). We therefore believe that the notable number of adaptive Ty1/2 insertions (120 insertions over four recurrently mutated loci -Table 1) best reflects an increase in Ty1/2 mutagenesis overall, rather than a selection bias for Ty mutations. This is further supported by a retrotransposition reporter assay, which shows an increase in Ty1 activity specifically under nitrogen starvation, but not glucose (S5 Fig and Methods). It remains to be investigated whether this activity is due to increased transcription or reinsertion, or if there are differences in Ty1/2 subclass activity, or specific Ty1/2 donor elements giving rise to the novel insertions.

To assess Ty1 activity in yeast experiencing different nutrient conditions, we used a modified version of a plasmid-based Ty1 reporter created previously [78]. To assess Ty1 activity induced by nitrogen limitation, we used the native Ty1 promoter from the insertion found at YPLWTy1-1. This Ty1 promoter was selected based on having recovered novel Ty insertions in evolved clones that originated from this donor locus. Strains bearing the Ty reporter plasmid were subjected to nitrogen limited media and glucose limited media to assess transposon activity in these conditions. We observe significantly more Ty activity from the reporter plasmid in nitrogen-limited conditions than in glucose limited conditions or in SC-Ura medium (S5 Fig).

## Directionality of novel Ty elements

Because Ty1 is known to have an enhancer that can increase expression of neighboring genes found co-oriented to novel Ty1 insertions, we investigated whether there was directional bias in novel Ty1 insertions [32–35]. We expect that if a novel insertion is disruptive, we would not see a preference for directionality of insertion, while insertions that are adaptive due to increased transcription from the Ty1 enhancer would show a bias for novel Ty1 insertions that

are co-oriented to the transcript of the adaptive target. In the case of all four nitrogen adaptive loci with multiple adaptive Ty1 insertions we did not observe a significant bias for strand-specific insertions (Fig 3 and S2 and S4 Files). Taken together, we infer that the majority of adaptive loci are likely functioning through the production of null alleles, or in the case of 3' *MEP1* insertions, through disruption of the antisense transcript, rather than through overactivation of transcription of these loci.

## Adaptive Ty1/2 insertions enriched in ORF regions

Ty elements preferentially target gene-poor regions and Ty1 in particular has been observed to most often land upstream of genes transcribed by RNA polymerase III, such as tRNA and snRNA genes [79]. In the hundreds of such Ty1/2 insertions that arose in nitrogen and glucose limitation we observed a periodicity of their insertion locations (S6 Fig) suggesting not only tRNA proximity bias, but also preference for nucleosome bound regions, consistent with previous observations [80]. We observed no preference for orientation of the Ty1/2 upstream of the tRNAs. A larger proportion of mutations in nitrogen neutral haploids were due to Ty1/2 element insertions (74%) relative to adaptive mutants (68%) (Fig 5A); however, the locations of adaptive Ty1/2 insertions show striking clusters at some of the nitrogen-adaptive loci (Figs 3 and 5), sites not typically preferred by Ty1.

## Discussion

Here we profiled adaptive mutations that arose in nitrogen-limited yeast cultures under conditions fluctuating between low-nitrogen and total nitrogen starvation [59], to better understand how selection regime shapes the molecular mechanisms of evolutionary adaptation. We observed that frequent Ty1/2 retrotransposition and microhomology-mediated recombination events are responsible for a large number of adaptive mutations. This is in contrast to glucose limitation under similarly fluctuating conditions, where self-diploidization and SNPs/short indels gave rise to most adaptive lineages [3,60]. Adaptive clones that arise to high frequency early in the evolution under glucose limitation show larger fitness effects than the respective clones under nitrogen limitation, accounting for the difference in the population dynamics [59]. Whether this reflects the altered spectrum of mutational types or is simply a result in the difference of strength of selection and distance from the fitness optimum is unclear, though these possibilities are not mutually exclusive.

In contrast to our observations in serial transfer experiments under nitrogen limitation, increased Ty activity is not a frequent mechanism of adaptation under nitrogen-limited continuous culture conditions, where cells experience a nutrient poor condition but never experience total starvation [55,58]. Instead, copy number variation of nitrogen transporters most often drives adaptation under continuous low nitrogen [55]. This adaptive mechanism bears resemblance to the means of adaptation observed in glucose-limited chemostats, where the copy number of *HXT* hexose transporters is often amplified, facilitating increased glucose uptake [42–44]. This suggests that despite different metabolite restrictions, chemostat evolution predominantly favors adaptation via increased abundance of transporters for the limiting nutrient, indicating these mutations arise most often and/or are the most fit. Similarly, in chemostat sulfate limitation amplification of *SUL* transporters is observed to be the most common adaptation [42,81,82]. Taken together these results suggest that an adaptive 'winning strategy' under chemostat conditions favors increases in transporter abundance, regardless of the specific nutrient restriction. While these amplifications are not the sole means of adaptation, the prevalence of these amplifications reflects both how easy these changes are to achieve (relative frequency of mutation type) and how beneficial (degree of fitness increase) these changes are

specifically in the low, but not zero, nutrient availability conditions of a chemostat. This suggests that under fluctuating conditions the period during the growth cycle in which additional transporters might be beneficial is insufficient to drive them to high frequency under such conditions–indeed, such amplifications may be deleterious at the beginning of a growth cycle, when the medium is replete with the eventually limiting nutrient [83].

Under serial transfer conditions, nutrient availability fluctuates and cells may even experience total starvation for the limiting nutrient. Previous work from our lab and others shows that under fluctuating glucose limitation self-diploidization and SNPs affecting the TOR and Ras pathways arise to high frequency early during such evolutions [3]. These mutations improve fermentation performance and also drive adaptation through decreasing time in lag phase upon reintroduction of fresh glucose [45]. By contrast, here we find that Ty1/2 and microhomology-facilitated mutations contribute substantially (though not exclusively) to the suite of adaptive mutations observed under nitrogen-limited serial transfer. This suggests two things: first, unlike chemostats, where the common 'strategy' for adaptation is frequently transporter copy number amplification, serial transfer does not initially favor a specific class of genomic changes irrespectively of the restrictive nutrient. Second, stably nitrogen poor conditions give rise to different adaptive 'routes' than fluctuating conditions where cells transiently experience total nitrogen starvation. This may be due, at least in part, to some mutagenic mechanisms, such as retrotransposon activity, being more or less active under nitrogen poor vs. starving conditions.

The significant increase of Ty1/2 insertions under fluctuating nitrogen-starvation conditions in particular, as compared to glucose limitation and chemostat conditions, raises the question as to what drives this increase, specifically in Ty1 activity. Ty1 derepression could be associated with host cell transcription changes due to stress, accumulation of other mutations [84] or even disruption of the Ty1 elements themselves, some of which restrict other subclasses of Ty1 [85]. Ty1 is known to contain a self-limiting restriction factor (p22) that facilitates copy number control within the host cell; it will be interesting to further explore how host nutrient starvation sensing interacts with this restriction pathway [86–89].

The increase in novel Ty insertions could also be due to increases in Ty reinsertion, or frequency of Ty reimport into the nucleus, rather than transcriptional activation. Studies determining at which step(s) of the Ty1/2 life cycle nitrogen starvation shapes Ty1/2 activity would be needed to determine the specific mechanism underlying the increase in retrotransposon insertions. Others have shown that Ty1 activity is sometimes due to an increase in transcription at specific Ty1 loci, during both severe adenine starvation and when environmental conditions trigger the invasive growth pathways—which includes nitrogen starvation [21,23–25,31,76,90–95]. It has also been shown that this transcriptional increase is not equal amongst Ty elements, and that specific retrotransposons were activated more than others. If the observed increase in novel Ty1/2 insertions in our results is also due to an increase in Ty1/2 transcription from a specific subset of originating loci (possibly the same subset as in [76]) it will be interesting to understand the mechanism underlying this donor element-specific activation.

The observed repertoire of beneficial mutations reflects both the underlying rate at which such mutations arise, as well as their resulting fitness. As such, we find it notable that some mutational mechanisms appear to be differentially active under certain environmental conditions, such as previously observed temperature control of Ty1 [96–98] and is the case with Ty1/2 activity under fluctuating nitrogen starvation here. This association of environmental stress with increase in Ty1/2 activity could also be a mechanism for stress induced mutagenesis [10,99]. If cells experience increased mutation rates while under specific stress conditions, but not under regular growth conditions, this could improve the overall population's chances of

survival. Host populations would sample greater genetic and phenotypic variation while experiencing conditions they might not otherwise survive, without the mutational burden during times of high fitness. A Ty insertion that is primarily silenced, but then active during extreme nutrient starvation could provide additional host-adaptive potential. Indeed, the role of retrotransposons in host evolvability is important: global Ty derepression could reflect host-parasite coevolution towards a less parasitic lifestyle, resulting in minimal host cost and maximized potential for survival of both, especially under detrimental environmental conditions [100]. If some strains are poised to "turn on" mutagenesis under otherwise deadly stress conditions, selection could favor these backgrounds; further evolutionary analysis of Ty insertions associated with starvation-induced activity could reveal one facet of selection shaping host-retrotransposon coevolution [23,24,31,101].

Alternatively, rather than being beneficial under stress and the trait being selected for, Ty activity may not have been strongly selected against in starvation conditions or may be impacted by historical survivor bias. Although competent under lab conditions, Ty elements rarely "hop" [102], so perhaps the correlation of Ty1/2 activity to stress response and environmental fluctuations is somewhat analogous to the meiosis-linked transposable element hopping in metazoans, where it is thought to minimize somatic damage and maximize spread in sexual populations [103]. Maybe low Ty activity was selected for during favorable yeast growth conditions, and the relative increase in activity reflects a lack of historical negative selection under nitrogen starvation. If this is the case, it would be interesting to better understand the mechanistic basis of why increased Ty1/2 mobilization occurs in total nitrogen starvation but not under constant low nitrogen conditions. Exploring the activity of mobile elements in other yeast species, including those with different lifestyles, such as pathogens, could provide further insight into the complex relationship between host and retrotransposon coevolution [104].

While Ty1/2 activity was notably increased under the evolution conditions we studied here, there were several other mutational mechanisms that were active and gave rise to multiple adaptive mutations. Notably, microhomology-mediated changes gave rise to gene conversion, duplication and deletion events. There are several mechanisms that can result in structural changes at sites of microhomology, including recombination and end-joining mechanisms, but also DNA replication-based mechanisms, such as fork stalling and template switching, or microhomology-mediated break-induced replication [105]. Like Ty1/2 activation, these mechanisms may occur more frequently than point mutations in the genome, and because of homologous sequences, can impact some regions of the genome more frequently than others. Homology-rich regions could be regions of instability relative to other parts of the genome, shaping host evolvability at those sites in particular and potentially minimizing deleterious effects elsewhere in the genome.

Several of the adaptive alleles we identified would likely have been missed in a classic loss-of-function genetic screen, including Ty-derived and microhomology alleles, but also some SNPs, such as the putative gain-of-function missense mutations in the *MEP* genes. In other cases, we identified adaptive alleles that we predict are null mutants, and that would be (and in some cases have been) identified in genetic screens. By taking an experimental evolution approach we also learn about the competitive fitness of these alleles–information that is not generated in most genetic screens. This emphasizes both the utility of experimental evolution for understanding the genetics of phenotypes of interest, but also how complementary this approach is to classical genetics.

When exploring a trait of interest, experimental evolution can both expand upon and reinforce lessons learned through other approaches. Natural variation in extant strains and species can help clarify what *has* happened in naturally evolving populations, but the context of those changes and their fitness consequences may not always be clear. Experimental evolution can

leverage large populations and controlled selective pressure to understand how genomes *can* evolve, what mechanisms give rise to novel alleles, and the relative fitness of those alleles. The alleles sampled will be shaped by the likelihood of their occurrence: the frequency is driven by the mutation mechanisms active in those cells, which can be shaped by the cells' environment, and the 'best' alleles will be those that provide the biggest fitness gains under the specific selective pressure. Lower fitness adaptive mutations can be well represented particularly if they arise more readily (such as autodiploids), or if they occur early in an evolution and secondary adaptive mutations are then also acquired. Often in genetic analyses, emphasis is placed on coding SNPs. While this class of mutation undeniably shapes how genomes evolve and organisms adapt, there are many other mutational mechanisms driving biological innovation. Studying the full complement of mutational mechanisms, and the spectrum of alleles that they create, provides a deeper understanding of important biological processes.

## Methods

### Evolutions media conditions

Nitrogen limited evolutions were previously described in [59]. Briefly, the ammonium-limited medium used was 5x Delft with 0.04% ammonium sulfate and 4% glucose. Glucose limited evolutions used 5x Delft with 4% ammonium sulfate and 1.5% glucose [60]. In both cases the limiting nutrient was previously confirmed to be restrictive, where addition of additional ammonium sulfate (up to 0.15%) or glucose (up to 4%) facilitated additional rounds of growth.

### Isolation and sequencing of evolved clones

Isolation of clones analyzed in this work were described previously [3,59]. Briefly, we isolated evolved clones from nitrogen- and glucose-limited evolutions from generation 192 and generation 88, respectively [3,59]. These generations were selected based on population dynamics of barcode frequency in the original evolutions; generation times were chosen late enough such that there was sufficient adaptation, yet early enough such that there would be expected to be few adaptive mutations per adaptive clone (see Blundell et al. 2019). Frozen aliquots from the appropriate time point were suspended in water and plated on SC–Ura plates. Single colonies were picked in unbiased fashion, suspended in 15 μL of water, and aliquots were saved for barcode PCR, ploidy determination (using a medium throughput benomyl sensitivity test [3]), and for storage in 25% glycerol at -80˚C. Haploid clones from independent lineages with fitness (as estimated from barcode trajectories during the nitrogen-limited evolution experiment) over 0 and self-diploidized isolates with fitness over 0.03, as well as 12 neutral clones were whole genome sequenced as described [59].

### Fitness remeasurement experiments

The 345 sequenced clones isolated from the nitrogen-limited evolution were pooled with 48 strains that were designated as having neutral fitness based on previous experiments [3]. The barcoded pool was then mixed with 90% unbarcoded wild-type (to minimize the change in population mean fitness during the experiment) and propagated via serial transfer for four transfers in either glucose-limited or nitrogen-limited media (as previously [59,60]). Barcodes were sequenced at each transfer, and fitness of clones was estimated from their barcode frequencies as described [59]. Fitness effects >0.01 were considered above the threshold of detection for beneficial fitness for these experiments as that is more than twice the standard deviation between technical replicate fitness remeasurements performed on the same strains.

## Multiplexed barcode sequencing

To determine individual barcodes in the selected clones, 5 μL of the suspension were lysed in 30 μL of lysis buffer (1 mg/mL Lyticase (Sigma L4025-250KU), 0.45% Tween20, 0.45% Igepal CA-630, 50mM KCl, 10mM Tris-HCl pH 8.3 (adapted from [106]). 2.5 μL of lysed cells were amplified in a 25 μL PCR reaction with 12.5 μL of 2xOneTaq mix, 5 μM of Forward (F(n)) and Reverse (R(n)) multiplexing primers (S1 Table) and water as follows: 94˚C x 2', 35 cycles of 94˚C x 30", 48˚C x 30", 68˚C x 30". Products from up to 6 plates were mixed together and run in one well of a 2% E-Gel SizeSelect II Agarose gel (Thermo Fisher Scientific, G661012) to purify 250bp amplicon. Purified amplicons were diluted 1:500 with water, and 2 μL were amplified in 25 μL reactions containing 1μM primers PE1 and PE1 with Phusion polymerase (NEB, M0530) as follows: 98˚C x 30", 12 cycles of 98˚C x 10", 65˚C x 30", 72˚C x 30". Final PCR products were purified with Qiagen PCR purification columns (Qiagen 28104), quantitated and mixed.

52 plates of clones were multiplexed and barcodes were sequenced in two paired-end 300bp MiSeq runs.

## Whole genome sequencing

DNA for whole genome sequencing was prepared using YeaStar Genomic DNA kit (ZymoResearch D2002). Sequencing libraries were prepared using the Nextera kit (Illumina FC-121-1031 and FC-121-1012) as described in [107], starting with 5–10 ng of genomic DNA. Resulting libraries from each 96-well plate were pooled at an equal volume. Pooled libraries were analyzed on the Qubit and Bioanalyzer platforms and sequenced paired-end 100bp on HiSeq 2000 (one lane per 96 clone pool).

## Variant and Ty insertion calling using CLC Genomics Workbench 11

Short reads from whole genome sequencing were trimmed to remove the adapter sequence and mapped (match score 1, mismatch cost 2, Linear gap cost: insertion 3, deletion 3, length fraction 0.5, similarity fraction 0.8) to *S. cerevisiae* reference genome (R64-2-1_20150113). SNPs and indels were called ignoring broken pairs with 85% minimal frequency in haploids and 35% minimal frequency in diploid strains, with base quality filter 5/20/15, direction frequency filter 5% and significance 1%, and read position filter 1%. Structural variation was called using InDels and Structural Variants identifying novel breakpoints. Breakpoints output from the InDels and Structural Variants calling (using the default settings) were used to identify Ty insertions (maximum 15% mapped perfectly for haploids and maximum 70% mapped perfectly for diploids). Breaks were filtered for unaligned sequences specific for LTRs and verified visually in the CLC Genomics Workbench browser.

## Identification of Ty insertions using RelocaTE2

Raw sequencing reads were trimmed using Cutadapt version 3.4 [108], and pairs of reads shorter than 60bp after trimming were removed. Mitochondrial reads were also removed. McClintock v2.0.0 [109] was run using default parameters. The *S. cerevisiae* reference genome file was downloaded from SGD [110]. Of the suite of tools available through McClintock we found that RelocaTE2 [111] performed best against a subset of manually curated Ty insertions, so RelocaTE2 outputs were used for further analysis. Insertion locations given in the RelocaTE2 output file were then mapped back to the S288C reference genome to determine what genetic feature the insertion was in ('gene', 'tRNA_gene', 'snoRNA_gene', 'snRNA_gene', 'long_terminal_repeat', 'transposable_element_gene'). The distance from each insertion

location to the closest downstream tRNA was calculated. Alignments at all RelocaTE2 candidate Ty insertions were visually inspected and were used to curate the insertions.

All variants from both CLC Bio and relocaTE2 are available in S1, S2, S3, and S4 Files.

## Mutant strain re-barcoding

Evolved barcoded strains containing mutated alleles of *gat1*, *par32*, *mep1*, *mep2*, and *mep3* were crossed to GSY5936-5940 to replace the double barcode with the barcode landing pad able to accept new barcoding. Multiple MATalpha ura- segregants from the crosses were combined and transformed with pBAR3-L1 to generate double barcodes as described [60]; multiple transformants were used to collect independently barcoded genotypes subsequently included in fitness remeasurements along with the original evolved strains.

## Fitness re-measurement

To re-measure fitness, barcoded strains were grown in 100 μL of YPD in microtiter plates and pooled. Pools were pre-grown in M14 or M3 medium [60], mixed with 10 volumes of culture of GSY5929 (wild type strain with digestible barcode) also pre-grown in M14 or M3 medium (time point 0 was taken right after mixing) and then passaged in triplicate through four batch transfers (400 μL into 100 mL of fresh M14 or M3 medium). 1.5 mL of culture grown for 48 hours (time points 1, 2, 3 and 4) were used for DNA preparation using the YeaStar Genomic DNA kit (ZymoResearch D2002).

75 ng of genomic DNA were used for PCR amplification in two 50 μL reactions with 25 μL 2xQ5 mix (NEB M0492), 2.5μM of each forward (FOS) and reverse (ROS) primers (S2 Table) and water (to 50μl) for 22 cycles (care was taken to not overamplify the libraries). After amplification DNA was digested with ApaL1 enzyme overnight, run on a gel to select uncut PCR fragments (approximately 300bp) and purified from the gel using the QIAquick Gel Extraction Kit (Qiagen 28704). Individual libraries were quantified, mixed equimolarly and sequenced in paired-end 300bp runs on MiSeq platform. Sequencing results were demultiplexed and resulting barcode frequencies were used to calculate the fitness as in [59].

To measure fitness of the backcrossed and individually barcoded derivatives of selected adapted clones, the strains were pooled (including the parental strains remeasured above) and mixed with the GSY145 (wild type strain without an amplifiable barcode). After four batch transfers in M14 medium cells were collected, DNA was prepared and barcodes were amplified as above, except that individual PCR reactions were not digested, but instead were gel size selected in pools of six. Care was taken to not overamplify the libraries adjusting the cycle number for amplification.

## Quantitation of transposon induction

To quantitate the induction of a Ty1 reporter construct in various media, we transformed a BY4741 derivative strain (MATalpha *his*3Δ *ura*3Δ) with plasmid pGS234, which was created by replacing the promoter containing XhoI fragment from pGTy1mhis3-AI with a 0.5kb XhoI fragment containing the promoter from chromosomal location of YPLWTy1-1. To induce the transposition, transformants were grown in SC–Ura and then 20 μL of overnight cell culture were added to 5 mL of SC–Ura, M3 or M14 media (the latter two were supplemented with histidine to a final concentration of 0.02 mg/mL). After 48 hours incubation in a roller drum at 30˚C 1.5 mL of cell suspension were pelleted and plated onto SC–His plates. Colonies were counted after 3 days of growth at 30˚C. YEp24 plasmid was used as a control (no His+ colonies were ever produced by those transformants). Cell concentration in cultures was monitored to make sure similar cell numbers were plated after 48 hours induction.

For the fluctuation test, one strain containing pGS234 was pre-grown in SC–Ura and then 20 μL of overnight culture were added to 16 tubes containing 5 mL of M3 or M14 supplemented with histidine as above. After 48 hours of growth 1.5 mL of cells from each tube were harvested and plated onto SC–His plates.

## Supporting information

**S1 Fig. Barcode trajectories during pooled fitness remeasurement experiments, A) in nitrogen limiting conditions, and B) in glucose limiting conditions.** A subset of known neutral lineages is represented by dotted lines. Lineages in red have an estimated fitness >0.01, in grey between -0.01 and +0.01, and in blue < -0.01.
(PDF)

**S2 Fig. Fitness of isolated clones is reproducible across biological replicates.** Estimated fitness between replicates for each clone under A) nitrogen limitation, and B) glucose limitation.
(PDF)

**S3 Fig. Locations of *MEP* mutations and antisense transcript at the *MEP1* locus.** A) Locations of Mep mutations in a multiple alignment; the green highlighted residue identifies which gene had that mutation, while the red highlighted residues were mutations observed previously [4], and B) transcription at *MEP1* locus indicates the presence of an antisense transcript downstream of *MEP1*.
(PDF)

**S4 Fig. Dynamics of increase in frequency of lineages with adaptive mutations in specific genes.** A) *PAR32* (red), B) *GAT1* (purple), C) *MEP1* missense mutations (light blue and burgundy), D) *MEP1* Ty1/2 insertions (green), E) *FCY2* (yellow), and F) fitness and lineage size as a function of establishment time (color-coded the same way as panels A-E). Black lines in A-E denote neutral lineages.
(PDF)

**S5 Fig. Increased Ty transposition in Nitrogen-limited media during batch culture.** (A) Bars represent average of three WT strains with a transposon reporter plasmid; each value is the number of colonies on SC-His medium with each His+ colony representing independent Ty transposition events. Strains were grown in SC-Ura and then shifted to M14, M3 or SC-Ura as a control for 48 hours and plated on SC-His plates. (B) One WT strain with pGS234 was subjected to a fluctuation test (16x 5ml tubes) in M14 and M3 media. Each dot represents the number of colonies on each SC-His plate. Kruskal-Wallis chi-squared = 23.341, df = 1, p-value = 1.357e-06.
(PDF)

**S6 Fig. Insertion locations of Ty1/2 elements upstream of tRNA genes in A) glucose limitation and B) nitrogen limitation.** New Ty1/2 element insertions typically prefer to land upstream of RNA Pol III transcribed elements, such as tRNAs [75, 79, 112].
(PDF)

**S1 Table. Primers used for Multiplexed barcode sequencing.**
(DOCX)

**S2 Table. Primers used for Barcode Amplification for Fitness Remeasurement.**
(DOCX)

**S1 File. All Variants identified in Nitrogen evolved clones.**
(XLSX)

**S2 File. Annotated output from RelocaTE2 for curated insertions from analyzing clones from Nitrogen evolution.**
(XLSX)

**S3 File. CLC Bio output for putative Ty insertion events in from analyzing clones from nitrogen evolution.**
(XLSX)

**S4 File. Annotated output from RelocaTE2 for curated insertions from analyzing clones from [3].**
(XLSX)

# Acknowledgments

The authors wish to thank Jamie Blundell for help with data analysis, and David Gresham, Frank Rosenzweig, and members of the Sherlock lab for helpful comments and discussions.

# Author Contributions

**Conceptualization:** Michelle Hays, Gavin Sherlock.

**Data curation:** Katja Schwartz, Danica T. Schmidtke, Gavin Sherlock.

**Formal analysis:** Danica T. Schmidtke, Gavin Sherlock.

**Funding acquisition:** Gavin Sherlock.

**Investigation:** Katja Schwartz, Dimitra Aggeli.

**Methodology:** Katja Schwartz.

**Project administration:** Gavin Sherlock.

**Software:** Katja Schwartz, Gavin Sherlock.

**Supervision:** Gavin Sherlock.

**Writing – original draft:** Michelle Hays, Gavin Sherlock.

**Writing – review & editing:** Michelle Hays, Katja Schwartz, Dimitra Aggeli, Gavin Sherlock.

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
