## [Decision Letter · Decision Letter 0]

20 Mar 2023

Dear Dr Sherlock,

Thank you very much for submitting your Research Article entitled 'Paths to adaptation under fluctuating nitrogen starvation: The spectrum of adaptive mutations in Saccharomyces cerevisiae is shaped by transposons and microhomology-mediated recombination' to PLOS Genetics via Review Commons.

The manuscript was fully evaluated at the editorial level and by independent peer reviewers. The reviewers particularly appreciated the attention given to the previous round of reviews. However, one important point has not yet been addressed. As indicated by Reviewer 3, the orientation of the Ty insertions must be provided. There are a few additional comments that you may choose to address in the introduction or discussion.

We therefore ask you to modify the manuscript according to the review recommendations. Your revisions should address the specific points made by each reviewer.

Yours sincerely,

Geraldine Butler

Section Editor

PLOS Genetics

Geraldine Butler

Section Editor

PLOS Genetics

Reviewer's Responses to Questions

**Comments to the Authors:**

Reviewer #1: I already reviewed this paper (Reviewer 3) for Review Commons, so for my full review, I refer to that text.

I am happy with the author's response and changes to the manuscript. Apart from the changes in response to my own review, I think that the authors also responded very well to the (minor) comments of the other reviewers, in particular the relation between transposon activity and selection regime is a very interesting addition that may merit a bit more attention in the discussion and perhaps even the abstract since it goes to the heart of the paper's main goal (investigating if and how selection regime influences the molecular mechanisms of evolutionary adaptation).

kevin verstrepen

Reviewer #2: My concerns have been addressed.

Reviewer #3: Rereview of Hays et al.

First things first. I love this paper and I think it’s super interesting. The new data not included in the revision are interesting for sure. They would add some heat to this paper! The authors have improved the paper but have utterly failed on one aspect (I regret not having been more forceful about this on the first round!) and it really has to be fixed.

Major critique:

1. One of the types of researchers who are going to be eager to read this paper is the retrotransposonologists. And the authors’ ignoring the recommendation to provide orientation information is simply unacceptable. When you describe a TE insertion it is one of the most fundamental aspects, as it governs the likelihood that adjacent gene sequences will be newly transcribed. As shown by Valerie Williamson, Stuart Scherer and a legion of other Ty1-ologists, there in an enhancer near the “left end” of the Ty1 and it is a potent activator of flanking gene expression! That’s just one of the reasons why this info is important in deducing the mechanisms involved in the adaptive and nonadaptive insertions.

Thus, and I know this is painful because your team did not do it up front, but you really have to go back and update your figures and tables to provide the orientation information. At MINIMUM, the following Figures and tables need to be updated to provide orientation information for each Ty element described: Most especially Figure 3 (orientation relative to target gene transcription), Figure S5 (orientation relative to target gene transcription) and all supplemental Tables (orientation relative to chromosome coordinates)

2. Similarly in some of the tables saying just “Ty” is not acceptable, each Ty can be ssigend to one of the families very easily since they have very different sequences. An exception to this is Ty1 and Ty2 which are similar (they are basically subfamilies), thus in these cases it is acceptable to list them as Ty1/Ty2

Minor critiques:

Below I list a whole slew of perhaps picayune critiques mostly of the transposon parts, please note they are aimed at making the intro and discussion part of your paper more accurate and hopefully also even more interesting.

2. Extremely picayune – but extremely annoying to reviewers and editors – you didn’t put page numbers or line numbers. You will have to hunt

3. You may want to change the word “transposons” to “retrotransposons” in the title to avoid confusion. Many use the word transposon to refer to DNA transposons only. Yeast has none of the latter. You also use the terms transposition when retrotransposition would be more accurate throughout the text.

4. Whenever possible do not use the term Ty which is collective. In almost all cases, use of the more specific term Ty1 (or Ty1/Ty2) is more specific and accurate

“including in Saccharomyces cerevisiae (many refs)”. Please add temperature regulation: Pacquin and Willliamson PMID: 17815421, Boeke et al. PMID: 3025601, Lawler et al PMID: 11932388

5. Table 1. Definitely specify Ty type – I expect they are all Ty1/2; ideally also specify orientation here. I am betting money that the MEP insertions will be interesting (i.e. non-random as to orientation). Same for figure 2 “Ty” allele, Fig. S4

6. MEP1 section: The discussion about possible insertional interference with the MEP1 antisense RNA seems very plausible. Please state whether or not there is any evidence for similar 3’ antisense RNAs for MEP2/3…

7. Incorrect sentence: “Others have observed Ty1 and Ty2 to be the most active classes of spontaneous Ty transposition (Curcio et al. 1990)”. This sentence really isn’t right. Curcio paper showed Ty1/2 were the only classes that could turn on gene expression. It turns out Ty3, 4 and 5 don’t have this ability as they lack the Ty1/2 enhancer. I would simply say “Ty1 and Ty2 are far and away the most abundant classes of Ty element (PMID 9582191) and are highly active in yeast (many refs).

8. “or even disruption of the Ty elements themselves, some of which restrict other subclasses of Ty (Czaja et al. 2020)”. This should be changed to Ty1 (twice).

9. “it will be interesting to understand the mechanism underlying this insertion-specific activation.” To avoid confusion, when you are talking about pre-existing elements in the genome, as in this case, use the term “donor element-specific activation”.

10. “as is the case with Ty activity under fluctuating nitrogen starvation”. This is a golden opportunity to reference temperature control of Ty1 hopping.

11. Suggest change to “If cells experience increased mutation rates while under [specific] stress conditions”

12. Para that begins “Alternatively, rather than …”. Really worth mentioning here that in the wild, yeast likely overwinters in the guts of wasps. PMID: 22847440

13. Methods: There is no such medium as SD–Ura, it is SC–Ura. Also, please be sure to use the minus sign and not the hyphen in this context!

14. Supp table 1. I believe that in column F, this is the authors’ valiant attempt to identify the donor element that gave rise to the insertion. I think this is a dangerous game for two reasons. 1) Many Ty1 copies are identical in sequence and thus cannot be tracked. 2) Are you really usre that all the Ty1 sequences in the proengitor strain you worked with are identical in sequence aqnd position to the SGD reference sequence, which I expect, is what was used to assign the Ty1 sequences? Probably better to be agnostic on this. Also, What do the asterisks mean. Table needs a legend

**Have all data underlying the figures and results presented in the manuscript been provided?**

Reviewer #1: Yes

Reviewer #2: Yes

Reviewer #3: Yes

PLOS authors have the option to publish the peer review history of their article (what does this mean?). If published, this will include your full peer review and any attached files.

Reviewer #1: **Yes: **Kevin Verstrepen

Reviewer #2: No

Reviewer #3: No

---

## [Editor Report · Decision Letter 1]

14 Apr 2023

Dear Dr Sherlock,

We are pleased to inform you that your manuscript entitled "Paths to adaptation under fluctuating nitrogen starvation: The spectrum of adaptive mutations in Saccharomyces cerevisiae is shaped by retrotransposons and microhomology-mediated recombination" has been editorially accepted for publication in PLOS Genetics. Congratulations!

Yours sincerely,

Geraldine Butler

Section Editor

PLOS Genetics

Geraldine Butler

Section Editor

PLOS Genetics

Comments from the reviewers (if applicable):

**Data Deposition**

http://datadryad.org/submit?journalID=pgenetics&manu=PGENETICS-D-23-00217R1

**Press Queries**

---

## [Editor Report · Acceptance letter]

11 May 2023

PGENETICS-D-23-00217R1 

Paths to adaptation under fluctuating nitrogen starvation: The spectrum of adaptive mutations in Saccharomyces cerevisiae is shaped by retrotransposons and microhomology-mediated recombination 

Dear Dr Sherlock, 

We are pleased to inform you that your manuscript entitled "Paths to adaptation under fluctuating nitrogen starvation: The spectrum of adaptive mutations in Saccharomyces cerevisiae is shaped by retrotransposons and microhomology-mediated recombination" has been formally accepted for publication in PLOS Genetics! Your manuscript is now with our production department and you will be notified of the publication date in due course.

With kind regards,

Timea Kemeri-Szekernyes

PLOS Genetics

On behalf of:
